# Neuroprotective Effect of Vascular Endothelial Growth Factor on Motoneurons of the Oculomotor System

**DOI:** 10.3390/ijms22020814

**Published:** 2021-01-15

**Authors:** Silvia Silva-Hucha, Angel M. Pastor, Sara Morcuende

**Affiliations:** Departamento de Fisiología, Facultad de Biología, Universidad de Sevilla, 41012 Sevilla, Spain; silvia_sh88@hotmail.com (S.S.-H.); ampastor@us.es (A.M.P.)

**Keywords:** VEGF, oculomotor system, trophic factors, motoneurons, neurodegeneration, axotomy, amyotrophic lateral sclerosis

## Abstract

Vascular endothelial growth factor (VEGF) was initially characterized as a potent angiogenic factor based on its activity on the vascular system. However, it is now well established that VEGF also plays a crucial role as a neuroprotective factor in the nervous system. A deficit of VEGF has been related to motoneuronal degeneration, such as that occurring in amyotrophic lateral sclerosis (ALS). Strikingly, motoneurons of the oculomotor system show lesser vulnerability to neurodegeneration in ALS compared to other motoneurons. These motoneurons presented higher amounts of VEGF and its receptor Flk-1 than other brainstem pools. That higher VEGF level could be due to an enhanced retrograde input from their target muscles, but it can also be produced by the motoneurons themselves and act in an autocrine way. By contrast, VEGF’s paracrine supply from the vicinity cells, such as glial cells, seems to represent a minor source of VEGF for brainstem motoneurons. In addition, ocular motoneurons experiment an increase in VEGF and Flk-1 level in response to axotomy, not observed in facial or hypoglossal motoneurons. Therefore, in this review, we summarize the differences in VEGF availability that could contribute to the higher resistance of extraocular motoneurons to injury and neurodegenerative diseases.

## 1. Vascular Endothelial Growth Factor (VEGF)

### 1.1. History

VEGF was initially described as an angiogenic factor and, consequently, it was named vascular permeability factor (VPF) for its role in inducing vascular permeability in tumor cells [1]. It was not until 1989 when the VEGF protein, whose molecular weight is approximately 45 kDa, was purified and sequenced, and it was definitively assigned the name of vascular endothelial growth factor [2,3].

It is well known that this factor is a highly specific mitogen for vascular endothelial cells, whose family consists of multiple cell signaling proteins involved in angiogenesis, lymphangiogenesis, vasodilation, and vascular leakage, among other functions [4]. In 2001, his essential role in motoneuronal protection was revealed for the first time [5], as will be discussed later in detail.

### 1.2. VEGF Family

Since the discovery of the first member of the VEGF family, known as VEGF-A, the family has continued to grow and is currently constituted by VEGF-A, VEGF-B [6,7], VEGF-C [8,9], VEGF-D [10,11], VEGF-E [12], VEGF-F [13] and placental growth factor (PlGF [14]).

VEGF-A stands out as the most studied VEGF family members because of its essential roles in neuroprotection. The gene encoding VEGF-A is located on chromosome 6p21.5, giving rise to three different isoforms (VEGF-A 121, VEGF-A 145, VEGF-A 165), which are differentiated by their molecular weight, solubility, biological functions, binding affinities to the components of the extracellular matrix, and tyrosine kinase receptor subtypes (RTKs) [15]. Furthermore, VEGF 165 is the predominant isoform in the central nervous system (CNS), where it acts as a protection factor by promoting the survival of motoneurons [16,17]. This vital role is the one that interests us and the one that we will develop on throughout this review.

On the other hand, VEGF-B is also expressed in the CNS and can regulate adult neurogenesis and even rescuing neurons from apoptosis, but with less vascular effects and worse neuroprotective function on motoneurons than VEGF-A [18]. It has recently been discovered that the presence of VEGF-B is neither necessary nor essential for the survival, maintenance, and development of motoneurons under normal physiological conditions [19,20]. Finally, mention should be made of the members VEGF-C and VEGF-D, which regulate lymphatic angiogenesis, and of VEGF-E, which is virally encoded and specifically expressed in the venom of the habu snake (*Trimeresurus flavoviridis)* [21].

### 1.3. Functions of VEGF

Throughout this review, we will refer to the VEGF-A 165 isoform, which is the one that exerts direct trophic and neuroprotective effects on many types of neural cells [22], including motoneurons [5,23], astroglia [24], microglia [25], hippocampal, dopaminergic, cortical, cerebellar, sympathetic neurons, and even muscle satellite cells [16,26,27,28,29,30,31,32].

Furthermore, this trophic factor plays a fundamental role in stimulating neurogenesis in both developing and adult CNS [16], promoting Schwann cells’ proliferation [27]. It also supports synaptic plasticity [33] and favors the growth, survival, differentiation, and migration of neuronal and glial cells [20,34]. Additionally, VEGF guarantees an optimal blood and glucose supply to the brain and spinal cord [5], protecting motoneurons from oxidative stress [35], hypoxia, hypoglycemia [23], and glutamate-mediated excitotoxicity [29,36,37,38,39].

VEGF is also a potent inducer of the blood-brain barrier’s interruption by increasing its permeability and favoring the supply of oxygen and nutrients to neurons [34,40]. This trophic factor is involved in vasculogenesis during embryological development and promoting angiogenesis in many pathological conditions, such as tumor growth, rheumatoid arthritis, psoriasis, and diabetic retinopathy [4].

### 1.4. VEGF Expression

Several factors have been found to upregulate VEGF mRNA expression, including tumor necrosis factor (TNF-α), platelet-derived growth factor (PDGF), interleukins, angiopoietins, and erythropoietins [41,42,43,44,45]. Another molecule that regulates the expression of VEGF is nitric oxide, which contributes to the processes of permeabilization of blood vessels and in vasodilation stimulated by VEGF [29,46,47].

However, one of the main and more robust regulators of VEGF expression is hypoxia [48]. VEGF mRNA has a half-life of 30–45 min under normoxic conditions, whereas the mRNA half-life is prolonged in hypoxia [49,50] and cells increase the production of the hypoxia-induced transcription factor 1 (HIF-1), a heterodimer consisting of three subunits (HIF-1α, HIF-1β, and HIF-3) [51,52]. HIF-1α and HIF-1β are produced continuously, but HIF-1α is highly labile in the presence of oxygen, so it degrades under aerobic conditions [53]. When the cell is in a hypoxic environment, HIF-1α persists and translocates to the nucleus, where it associates with HIF-1β and forms the HIF-1α/HIF-1β complex. This complex binds to the hypoxia response element (HRE) [54,55], whose transcriptional activation requires the recruitment of the CREB-binding protein, which is a transcriptional coactivator. Thus, as the 2019 Nobel Laureates in Medicine Kaelin, Ratcliff, and Semenza described, through this mechanism, cells perceive and adapt to changes in oxygen levels, modifying both their metabolism and physiological functions [52,56]. Therefore, transactivation of HRE by the HIF-1α/HIF-1β complex stimulates the gene expression of erythropoietin, glucose transporters, glycolytic enzymes, and VEGF [57,58], among others, the latter being in charge of promoting angiogenesis after binding to its specific receptors [5]. Little is known about the expression and function of HIF-3 [59].

## 2. VEGF Receptors

The biological activity of the VEGF family is mediated through binding to two classes of receptors: receptors with tyrosine kinase activity and receptors without tyrosine kinase activity. The first group consists of three structurally related receptors characterized by the presence of seven immunoglobulin-like domains in the extracellular region, a single transmembrane region, and an intracellular consensus tyrosine kinase sequence interrupted by a kinase insertion domain. These receptors are VEGR-1 (Flt-1), VEGFR-2 (KDR/Flk-1), and VEGFR-3 (Flt-4). On the other hand, the receptors without kinase activity are neuropilin-1 (NRP-1) and neuropilin-2 (NRP-2), which are also receptors for semaphorins [15,20] (Figure 1).

NRP-1 and 2 are expressed in different types of neurons [60,61] and play an essential role in regulating and developing the cardiovascular and nervous systems. Besides, they act as co-receptors for RTKs, presenting and improving VEGF binding to Flk-1 and promoting receptor phosphorylation and neurotrophic factor-mediated signal transduction [62,63].

Many studies indicate that both Flt-1 and Flk-1 activation could produce neuroprotection, but there are differences between the functions of both receptors [5,26,29,30]. Flk-1 predominates in neuronal and Schwann cells and is necessary for endothelial proliferation and migration, while Flt-1 is expressed mainly in vessels, astrocytes, and reactive microglia [29,64]. Besides, it has also been described that Flt-1 acts as a negative regulator for VEGF in endothelial cells, preventing its binding to Flk-1 [4] and that its functions and signaling properties may differ according to the stage of development of the animal, the cell type, and the binding ligand [65].

Unlike Flt-1, the Flk-1 receptor is an important survival promoter for endothelial and CNS cells, being the primary mediator of VEGF functions [4,66]. The main ligand of this Flk-1 receptor is the VEGF-A isoform, being the only one that triggers its auto-phosphorylation and final glycosylated form [4]. Therefore, the VEGF receptor that we will refer to throughout this review is Flk-1, which is expressed in motoneurons of the human spinal cord [40], mouse [5], rat [38] and is reduced in some patients with amyotrophic lateral sclerosis (ALS) [40]. Flk-1 overexpression in spinal motoneurons of the ALS SOD1 mouse model (with mutations in the gene encoding the antioxidant copper/zinc superoxide dismutase) has been shown to delay both neurodegeneration and disease onset [67], being the primary mediator of the neuroprotective and anti-excitotoxic effects of VEGF on motoneurons [37,38,68]. All these functions are performed by activating the phosphatidylinositol-3 kinase-AKT (PI3-K/Akt) pathway, involved in the processes of cell growth, proliferation, cell survival, and intracellular traffic, among others, in addition to regulating the entry of glucose to the cell through an insulin signaling cascade [69]. Furthermore, VEGF binding to the Flk-1 receptor also exerts a protective effect by suppressing the activation of the mitogen-activated protein kinase p38 (p38MAPK), a determining factor in the cell death pathway [16,30,38,70].

## 3. Effects of Low Levels of VEGF

The experiments carried out by Oosthuyse et al. in 2001 [5] were the first to suggest that VEGF acted as a neurotrophic factor at the CNS, as the reduction of VEGF function induced a specific degeneration of motoneurons in the adult mice. In these experiments, manipulation of the VEGF gene resulted in homozygous knock-in mice (VEGF^δ/δ^), in which the sequence of the hypoxia response element in the VEGF promoter region was removed. Consequently, these mice lost the ability to increase VEGF expression in a hypoxic situation. This alteration caused them severe muscle weakness due to the degeneration of the lower motoneurons, and they became progressively less mobile, with symptoms reminiscent of neuropathological signs of ALS. Although basal VEGF levels in muscles, heart, and fibroblasts were unaffected by removing the hypoxia response element, an overall 40% reduction of the neurotrophic factor was observed in neural tissue.

Those results obtained with the VEGF^δ/δ^ mice allowed linking for the first time a low level of VEGF with motoneuronal degeneration. Besides, mice resulting from the crossing of the SOD1 mutant with VEGF^δ/δ^ mice exhibited an even more drastic reduction in VEGF levels, thence a more severe degeneration of motoneurons and an earlier onset of symptoms of muscle weakness [17]. All these findings granted VEGF an unexpected neuroprotective role in the degenerative processes that accompany the pathogenesis of motoneurons, supporting the idea that motoneurons seem to be particularly sensitive to a low VEGF support [5,34,71]. Thus, a link was established between low levels of VEGF and neurodegeneration of motoneurons, such as occurs in ALS, and raised great expectation in VEGF as a possible candidate for ALS specific treatment.

### VEGF and ALS

ALS is characterized by being an adult neurodegenerative disease that causes progressive degeneration of motoneurons in the lower spinal cord, brainstem, and cortex. Consequently, it triggers astrogliosis, progressive atrophy of the skeletal musculature, and a reduction in voluntary movements, including those of the extremities and respiratory movements [72,73].

The disease affects five out of every 100,000 people worldwide, is progressive, and is generally fatal within 5 years after the onset of symptoms. 95% of cases are sporadic, and only 5% of patients have a family history, with a fifth of these caused by mutations in the SOD1 gene, located on chromosome 21 [15]. There are currently around 180 known mutations in SOD1 which are related to the pathogenesis of the disease [74], with the SOD1^G93A^ mutant mouse model being the most widely used, studied, and well-characterized showing symptoms similar to the disease [75], and to which we refer in this review as the SOD1 model.

It has been shown that the expression of VEGF and its Flk-1 receptor undergo significant downregulation in the motoneurons of the spinal cord of SOD1 mice [76]. These findings correlate with other studies where SOD1 mice were crossed with transgenic mice that overexpressed Flk-1, resulting in a delayed onset of motor impairment and degeneration of motoneurons, and prolonged survival [67]. Moreover, as indicated before, the coincidence of both SOD1 mutation and VEGF^δ/δ^ alteration produced an earlier and more severe motoneuronal degeneration [17].

Although the predominant hypothesis is that ALS is a disease of neural origin, some studies indicate that the disease involves a distal axonopathy and denervation of the neuromuscular junctions (NMJs) in the muscles of the extremities in the presymptomatic stage, that is, much before the loss of the motoneuron at the level of the spinal cord [77,78,79]. Several studies have shown that both the anterograde transport of VEGF and the intracerebroventricular infusion of this factor help protect and preserve the NMJs in a rat model SOD1 [67,80,81]. Likewise, experiments such as gene therapy of VEGF mediated by lentiviral vectors, or transplantation of stem cells that overexpress VEGF, have managed to significantly slow the progression of neurodegeneration, improving motor function and significantly prolonging the survival of motoneurons in the brainstem and the cervical and lumbar spinal cord [82,83]. All these findings give VEGF and its Flk-1 receptor an essential role in the treatment of motoneuronal diseases.

## 4. Neuroprotective Effect of VEGF

Among the possible pathogenic mechanisms linked to the degenerative neuronal process are oxidative stress, glutamate excitotoxicity, inflammation, mitochondrial and neurofilament dysfunction, protein aggregation, axonal transport abnormalities, and, ultimately, the activation of pathways that trigger apoptosis [84]. Numerous studies show the decisive neuroprotective role that VEGF plays in the CNS. These include experiments where an intracerebroventricular administration of VEGF stimulates neurogenesis in the adult hippocampus, promotes neurites growth, or provides greater protection to motoneurons [85,86]. Furthermore, it has also been shown that the retrograde transport of VEGF, after its intramuscular administration with lentiviral vectors, favors the survival of motoneurons [17,82]. At the same time, the supply of VEGF at the site of a spinal cord injury decreases lesion size, apoptosis levels, and retards neurodegeneration [36,87].

Two main hypotheses have been postulated to explain all of these VEGF effects. The first affirms that this factor promotes the vascular niche necessary for motoneurons to survive, and the second, that the binding of VEGF to Flk-1 promotes cell survival by blocking the process of apoptosis.

Another protective effect of VEGF is also due to the induction of the expression of the GluA2 subunit in AMPA receptors [64], which leads to a reduction in the entry of Ca^2+^ in neurons, which is a relevant mechanism involved in motoneuronal degeneration. All this makes VEGF and its Flk-1 receptor attractive candidates for evaluating its therapeutic potential in neurodegenerative disorders.

### 4.1. Anti-Apoptotic Effects of VEGF

One of the mechanisms by which the binding of the VEGF-A isoform to the Flk-1 receptor improves and promotes cell survival is by blocking the process of apoptosis through the expression of anti-apoptotic proteins, such as the members of the Bcl-2 family and the generation of neuronal progenitors in the nervous system [16,88,89].

It is well known that the binding of VEGF with Flk-1 directly activates the PI3-K/Akt intracellular signaling pathway. This activation consequently causes an inhibition of the phosphorylation of p38MAPK, which is an essential factor in the cell death pathway [38,64], and an increase of the expression of the anti-apoptotic proteins Bcl-2 and A1, conceding greater protection to motoneurons against excitotoxicity. That effect has been described in diverse models of neurodegeneration, including ALS [38,70].

### 4.2. Role of Excitotoxicity in Neurodegeneration and VEGF Protection

Glutamate is the major excitatory neurotransmitter in the mammalian CNS and is involved in many aspects of normal brain function. However, an excess in the synaptic transmission of glutamate leads to an over-activation of the different types of receptors for this amino acid, which causes a massive entry of Ca^2+^ in the neurons and triggers the uncontrolled activation of damaging processes that, eventually, produce the destruction of the membrane, neurodegeneration and cell death [90]. Indeed, glutamate-mediated excitotoxicity is considered the primary mechanism leading to a degeneration of motoneurons in various neurodegenerative disorders, including Parkinson’s disease, Alzheimer’s disease, and ALS [91,92,93].

Two broad categories of glutamate receptors are known: (i) ionotropic receptors, which are ligand-activated ion channels, and comprise *N*-methyl-D-aspartate (NMDA), α-amino-3-hydroxy-5-isoxazole propionate (AMPA), and kainate receptors; (ii) metabotropic receptors, which are associated to G proteins and coupled to the production of intracellular secondary messengers [94,95]. AMPA ionotropic receptors are heteromeric complexes composed of four subunits, GluA1-GluA4 (formerly GluR1-GluR4), with different combinations. The presence of the GluA2 subunit is known to decrease the permeability of these receptors to Ca^2+^ [95]. Therefore, the hyperactivation of AMPA receptors that lack this GluA2 subunit involves a massive entry of Ca^2+^ into the cell, which produces the activation of phospholipases, proteases, and endonucleases, inducing apoptotic or necrotic cell death, the production of reactive oxygen species (ROS) and a deficiency of mitochondrial function, with the consequent interruption of energy metabolism [90,96]. All of this generates neuronal degeneration, suggesting that the absence of the GluA2 subunit at AMPA receptors is a critical factor for the selective vulnerability of neurons to excitotoxicity.

Motoneurons are particularly susceptible to excitotoxicity due to their low expression of the GluA2 subunit [97,98]. Deficiency of this subunit exacerbates motoneuron degeneration in SOD1 mouse models, whose mutation is implicated in the accumulation of oxidative damage [99]. On top of that, many ALS patients have shown elevated glutamate levels in the cerebrospinal fluid, supporting the excitotoxic hypothesis of degeneration of the motoneurons [100]. Furthermore, glial cells that overexpress the SOD1 mutation are known to adversely affect the viability of spinal motoneurons by producing elevated levels of extracellular glutamate, leading to increased motoneuron degeneration and progressive paralysis [101].

Interestingly, the administration of exogenous recombinant VEGF is capable of preventing both excitotoxic neuronal death, induced by overactivation of AMPA receptors, and the consequent motor disorders [36]. That reduction in glutamate toxicity is mediated by the action of VEGF on PI3-K/Akt and MEK/ERK pathways [102]. Moreover, this neurotrophic factor has been shown, both in vitro and in vivo, to induce an increase in the expression of the GluA2 subunit in AMPA receptors [103], reducing the permeability to Ca^2+^ and, therefore, granting protection to motoneurons against excitotoxicity. Those experiments highlight the relevant role that VEGF plays in reducing excitotoxicity, making this factor an essential piece for the survival of motoneurons.

## 5. Selective Vulnerability of Motoneurons to Neurodegeneration

As stated above, motoneurons are particularly sensitive to excitotoxic neurodegeneration due, mainly, to their reduced capacity to blockade Ca^2+^ influx. However, a peculiarity of neurodegenerative processes is that specific neuronal populations show superior resistance to degeneration compared to other motor groups. In diseases such as ALS, motoneurons of some motor nuclei offer selective resistance and persist until the last stages of the disease, compared to other motoneurons that degenerate earlier [104]. Motoneurons of the oculomotor system are among those resistant populations [104,105], while motoneurons of the facial, hypoglossal, or trigeminal motor systems are vulnerable populations in the brainstem [106].

Likewise, between the possible differences that mark the selective vulnerability of the different motoneuronal groups is the differential expression of specific proteins. Several studies have shown that ocular motoneurons have a distinct transcriptional profile from other motoneurons in the expression of proteins related to synaptic transmission, including several glutamate and GABA receptor subunits, Ca^2+^-binding, ubiquitin-dependent proteolysis, mitochondrial function, or immune system processes [107,108,109].

Moreover, a greater expression of laminins, synaptophysin, and p75 receptor has been detected in muscle fibers of resistant motoneurons [110,111]. Thus, the differential expression of specific proteins and neurotrophic factors on the target muscles seems to influence the selective resistance of the different motor units against neurodegeneration and the deterioration of the NMJs, granting a vital role to the retrograde trophic contribution in motoneuronal survival.

## 6. Properties of Ocular Motoneurons

The ocular motoneurons present a series of morphological and functional characteristics that differentiate them from the rest of the motoneuronal populations. Several hypotheses have been proposed to explain the greater resistance of these cell populations to neurodegeneration.

Motoneurons of the ocular system show an extensive buffering capability of intracellular Ca^2+^ due to a greater expression of cytosolic Ca^2+^ binding proteins [112,113,114]. Overexpression of Ca^2+^ binding cytosolic proteins, such as calbindin D-28K (CaBP), calretinin (CR), and parvalbumin (PV), seem to give high protection to motoneurons [115,116,117,118]. Likewise, it has been observed that 85–100% of the motoneurons of the primate ocular motor nuclei contain PV, while only 20–30% of the neurons of the trigeminal, facial, and hypoglossal nuclei present it [109]. This PV distribution pattern coincides with the selective vulnerability between the brainstem motor nuclei [104]. Furthermore, additional experiments in SOD1 mice revealed that PV levels were significantly higher in ocular motoneurons compared to hypoglossal motoneurons [108].

Additionally, other studies in ALS models have related a low expression of the neuropeptide calcitonin gene-related peptide (CGRP), with higher resistance of motoneurons. Thus, CGRP could be a factor that promotes neuronal degeneration. Accordingly, motoneurons of the rat oculomotor system show lower expression of CGRP compared to other vulnerable motoneurons, such as facial or spinal ones, which would support these results [119,120]. However, these results have only been demonstrated in rats since a constitutive expression of CGRP has been observed in cats [121].

Notably, the extraocular motoneurons and the EOMs also express a higher proportion of the insulin-like growth factor 2 (IGF-2), which acts as a survival factor for motoneurons, and of its receptor IGF-1R, which mediates its survival effect [122,123]. Moreover, IGF-2 delivery to muscles preserved motoneurons and extended life-span in SOD mice [122]. It has also been shown that receptor α1 of the inhibitory neurotransmitter GABA-A (Gabra1) is preferably present in resistant motoneurons, such as ocular motoneurons, of symptomatic SOD1 mice and patients with end-stage ALS [107,108]. In contrast, vulnerable motoneurons show higher levels of GABA-A receptors α2 (Gabra2), dynein, and peripherin (intermediate neurofilament), which are involved in excitability and retrograde transport, which put these motoneurons at a higher risk [108]. Indeed, dysregulation in the dynein-dynactin or peripherin complex is well known to cause degeneration of the spinal motor neuron in mice due to faulty axonal transport [124,125], which is corroborated by the low levels of dynactin shown by spinal motoneurons in ALS patients [126]. Thus, the lower expression of dynein and peripherin is correlated with a lower vulnerability during neurodegenerative processes since retrograde transport is not affected [127]. Therefore, ocular motoneurons could continue receiving a correct trophic contribution from their target muscles, favoring the maintenance of their NMJs. 

Motoneurons present heterogenic neurotrophic dependence [128,129,130]. It is known that these nerve cells receive NGF, BDNF, and NT-3 from the muscle [131] and that the need for neurotrophic contribution, as well as the expression of the different RTKs, both in a control situation and after inducing a lesion, vary among the diverse populations of motoneurons [120,132,133,134,135]. Indeed, the adult rat spinal and cranial motoneurons are known to express the TrkB and TrkC receptors but lack the TrkA receptor [132,136,137,138,139]. However, another peculiarity of ocular motoneurons is that they express TrkA both in control and after axotomy in the adult [120,140]. This gives them greater efficiency in their response to NGF [120,141], which acts as a potent survival factor for axotomized neonatal motoneurons [142] and plays an essential synaptotrophic and functional role in axotomized motoneurons of the abducens nuclei [140].

Remarkably, recently it has been shown that motoneurons of the oculomotor nuclei present higher expression of VEGF and Flk-1 in the motoneuronal soma compared to other more vulnerable groups of motoneurons, such as the facial and hypoglossal [143]. Previous studies have also shown weak immunoreactivity for VEGF in hypoglossal and facial motoneurons in control rats [144]. The neuroprotective role of VEGF has been extensively exposed above and, therefore, could also contribute to the extended survival of ocular motoneurons in neurodegenerative diseases.

### 6.1. VEGF and FLK-1 Expression in the Oculomotor System

A high level of VEGF expression has been broadly related to neuronal survival. The expression of VEGF and Flk-1 is high during the embryonic stages but decreases during the adult state, being restricted to some areas of the adult CNS [145]. A higher basal level of VEGF and its receptor Flk-1 has been detected in ocular motoneurons compared to other brainstem motoneurons that are more vulnerable to neurodegeneration [143]. Thus, these oculomotor neurons could form one of these discrete CNS regions that retain the ability to express VEGF and Flk-1 after development.

Likewise, VEGF decreases the levels of pro-apoptotic proteins caspase-3, caspase-9, and Bax, and induces an increased expression of the anti-apoptotic protein Bcl-2 [146]. Therefore, the higher expression of VEGF observed in oculomotor neurons could yield a greater expression of anti-apoptotic proteins, which may be one of the reasons why these neurons show resistance against neurodegeneration. Furthermore, it has been observed that an increased expression of VEGF leads to a greater expression of its Flk-1 receptor [147], which correlates with the fact that a higher expression of Flk-1 was observed in those motoneurons that in turn expressed more VEGF [143].

Motoneurons are known to be especially susceptible to changes in Flk-1 expression, with a linear relationship between a lower expression of this RTK and a more significant loss of motoneurons. This occurs as a consequence of the blockade of the neuroprotective effect of VEGF, preventing activation of the PI3-K/Akt pathway, phospholipase C, and the p38MAPK protein [38]. These claims are supported by other studies where it was shown that the degeneration of spinal motoneurons could be delayed in transgenic SOD1 mice that overexpressed the Flk-1 receptor, thanks to the survival signals generated after VEGF binding [67]. All these findings give the Flk-1 receptor a key role in selective resistance that specific populations of motoneurons show against excitotoxic processes and neurodegeneration.

### 6.2. VEGF Sources to Ocular Motoneurons

The high level of VEGF found in the soma of extraocular motoneurons compared to the observed in other brainstem motoneurons [143] could be one more of the reasons for the lower vulnerability of this population to neurodegeneration. But, which is the origin of that higher VEGF content? Several possibilities could be considered: (i) VEGF could be synthesized by the motoneurons themselves and act as an autocrine source for extraocular motoneurons; (ii) it could reach motoneurons from surrounding cells, such as glial cells and endothelial cells on the blood vessels; (iii) VEGF could also come from the target muscles via retrograde to the innervating motoneurons (Figure 2).

#### 6.2.1. Via Autocrine

Motoneurons are known to synthesize trophic factors, including neurotrophins such as BDNF, NGF, and NT-3 [148,149]. It is well-known that they also express their receptors, allowing these motoneurons to receive and use the trophic factors as an autocrine source [141,149]. Their production has been shown to vary in response to diverse insults [120,150].

As aforementioned, ocular motoneurons can synthesize VEGF both in control situation and after injury [151], and they express VEGF receptors on their surface [143,151]. Therefore, it is possible that those motoneurons are also acting as an autocrine source of VEGF. Thus, the increased VEGF highlights the autocrine functions of the VEGF, as previously described in the CNS [22,152], this pathway being one of the essential vias of VEGF supply to ocular motoneurons [151].

#### 6.2.2. Via Paracrine

The high level of VEGF located in extraocular motoneurons could also indicate that VEGF is acting as a paracrine factor for the adjacent neurons. The fact that the presence of the Flk-1 receptor is increased in this pool of motoneurons allows them to receive higher amounts of the trophic factor from the neighboring cells. Thus, the upregulation of Flk-1 in the ocular motoneurons highlights the paracrine functions of VEGF [151,152].

Astrocytes are involved in almost all physiological processes that ensure the well-being of neurons [153,154]. Likewise, astrocytes also play a role in neurodegenerative processes since the selective deterioration of the glutamate transporter EAAT2 causes the extracellular accumulation of excitotoxic levels of this amino acid and an increase in the entry of Ca^2+^ in neurons [155]. ROS are believed to induce this oxidative disruption of glutamate transport and promote the spread of this damage, affecting motoneurons. Consequently, glutamate levels increase further, inducing more ROS in motoneurons and triggering a progressive cascade of selective motoneuronal injury, with consequent astrocytic and microglial activation [156]. On the other hand, the cells of the microglia are the specialized macrophages of the CNS [157]. They are an essential component of the inflammatory response to lesions and pathogens [158]. After an injury to the CNS they are the first glial cells to respond, producing pro-inflammatory mediators [159] and promoting the reaction of neurotoxic astrocytes [160].

However, the expression of VEGF driven by the glial cells surrounding brainstem motoneurons is low under control circumstances [151]. A low expression of both mRNA and VEGF protein in glial cells in a control situation was also previously described [161,162], ruling out the possible role of these neural cells as a paracrine source of VEGF to motoneurons at basal conditions. Therefore, astrocytes and microglia do not seem to be contributing to the differential expression of VEGF detected between oculomotor, facial, and hypoglossal motoneurons in a control situation. Nevertheless, glial cells have been reported to modify their VEGF expression under adverse conditions [25,163].

It is well-known that VEGF also acts as a growth factor for vascular endothelial cells forming the blood vessels [2], promoting vascular proliferation and permeability and therefore providing oxygen and nutrients to neurons, which contribute to their wellness. Administration of exogenous VEGF in the brainstem is not accompanied by either angiogenesis or a significantly increased vascular permeability around treated motoneurons [164]. Therefore, it could be assumed that the action of this factor on motoneuron survival was likely due to a direct effect on the motoneurons instead of an indirect effect due to increased blood perfusion. Besides, no differences were observed in the vascularization of these motor nuclei, neither in control nor after an injury [151,164].

Therefore, paracrine actions of VEGF do not seem to be crucial for the differences observed in resistance to degeneration between diverse pools of brainstem motoneurons.

#### 6.2.3. Retrograde Via

Three pairs of extraocular muscles (EOMs) are inserted around the eye, functioning as antagonistic to each other. These are: (i) the medial rectus and lateral rectus muscles, producing eye movements in the horizontal plane; (ii) the superior and inferior rectus muscles, in charge of vertical movements; and (iii) the superior and inferior oblique muscles, which mediate oblique movements [165].

EOMs are anatomically and functionally quite different from other muscles (reviewed in [166]). Most skeletal muscles exclusively have single innervation fibers (SIF), with a single axon forming part of the NMJs and constituting the motor unit, in which a motoneuron innervates 300–2000 muscle fibers (ratio 1: 300–2000). Furthermore, these SIF fibers have a high content of mitochondria and oxidative enzymes, which results in faster contractions. On the other hand, EOMs present a high percentage (20%) of fibers with multiple innervations (MIF), characterized by forming smaller motor units (1:5 ratio), with lower mitochondrial content and fewer oxidative enzymes, a relatively slow, graduated contraction [166,167]. This constitution favors a fine and precise muscular control, modulating the ocular movement and resulting in a more stable vision [168].

It is important to highlight that fast motor units degenerate before slow ones, due, at least in part, to the fact that the motoneurons that supply the slow contraction muscles can compensate the death of its neighboring motoneurons temporarily by generating compensatory axonal branches and reinnervation of the denervated muscle [166]. In contrast, motor-neuronal populations that exclusively present fibers with SIF innervation lose contact with their target muscles much earlier and, consequently, are more vulnerable to neurodegeneration. This resistance has been demonstrated in SOD1 mouse models, where the EOMs remain fully innervated in stages in which the limb muscles show deep denervation [108,169,170]. Therefore, the EOMs can maintain NMJs for a longer time, which leads to a greater retrograde trophic support from the EOMs to the projecting motoneurons.

EOMs have been shown to express VEGF and, therefore, are good candidates to intervene in trophic supply towards motoneurons [151]. Previous studies described the anterograde and retrograde transport of VEGF to neurons, the latter being crucial for maintaining the integrity and functionality of NMJs [82,171]. The importance of trophic supply to ocular motoneurons is also emphasized by the higher expression of BDNF, NGF, and NT-3 found in the EOMs, compared to the buccinator and tongue muscles, target muscles for facial and hypoglossal motoneurons, respectively [172], emphasizing the role of the retrograde pathway as a source of trophic factors.

Although the level of VEGF expression is similar in buccinator, tongue, and EOMs, there is a higher density of Flk-1 receptors in the pre-synaptic terminal of the EOMs compared to the muscles innervated by facial and hypoglossal motoneurons [151]. Previous studies have also shown the presence of the Flk-1 receptor at NMJs level of the abducens motoneurons, projecting by the abducens nerve towards the lateral rectus muscle [173]. These data support the idea that, although extraocular, facial, and hypoglossal muscle fibers were found to be positive for VEGF, not all target muscles appear to be acting to the same extent as the retrograde source of this factor towards the motoneurons that innervate them [151]. In this sense, the VEGF reaching the motoneurons of the ocular motor system through the retrograde pathway may have a more significant influence than the VEGF that comes to the facial or hypoglossal motoneurons.

All this evidence suggests that the retrograde function of VEGF is important and determinant for the survival of brainstem motoneurons.

### 6.3. Characteristics of the Ocular Motor System after Axotomy

#### 6.3.1. Regulation of Trophic Factors

Upregulation of VEGF expression seems to be a common phenomenon in response to a lesion since any injury to the CNS is known to trigger a hypoxia process involving VEGF expression [144,174]. These findings suggest that the upregulation of endogenous VEGF may be related to its neuroprotective role, so there would be a selective and preferential induction of this neurotrophic factor in some areas of the brain, which would allow greater binding of the ligand to its receptors and would provide greater trophic support [175]. In this sense, it is important to highlight that extraocular motoneurons suffer an increase in the expression of VEGF and Flk-1 in response to axotomy, an increase that was not observed in facial and hypoglossal motoneurons [151].

That increases in VEGF expression has been previously detected in response to ischemia or seizures in both neurons and glial cells of the hippocampus, thalamus, amygdala, and neocortex [175,176]. These results have great functional relevance since VEGF seems to maintain neuromuscular communication even during the denervation processes of the target and are consistent with the neuroprotective role that VEGF exerts on motoneurons, prolonging their survival and improving motor performance [17,82,177,178].

As discussed before, it is well known that the neuroprotective effects that VEGF plays on motoneurons are mediated by the Flk-1 receptor, which is involved in the release of growth factors [64] and mediates trophic functions [37,38,40]. Therefore, this evidence emphasizes the importance of the increased expression of Flk-1 observed in the ocular motoneurons in response to axotomy [151], this being one of the possible keys to the greater resistance shown by this population against neurodegeneration.

Previous studies showed changes in the expression of other trophic factors, such as neurotrophins, or their receptors in motoneurons after axotomy [120,132,179]. It has also been described that axotomized motoneurons experience a decrease in the expression of the protein acetylcholintransferase (ChAT) [120,132,164]. It is important to note that both the immunoreactivity of ChAT and the activity and mRNA of this enzyme are also markedly reduced in cases of ALS [180,181,182]. In fact, the motoneurons of the anterior horn of the spinal cord suffer a decrease in ChAT activity from early stages, compared to control neurons [183], which suggests that a reduction in ChAT expression is a specific and initial change in disease pathogenesis [184]. It is worth mentioning that the exogenous administration of VEGF at the site of the injury is capable of preventing the loss of the cholinergic phenotype in the axotomized motoneurons, which allows the ocular motoneurons to retain a neurotransmissive phenotype [164]. These studies emphasize the importance of the neuroprotective effect that VEGF has on motoneurons by maintaining its synaptic transmission capacity.

Therefore, all this evidence suggests that facing an injury to the CNS that involves the loss of the muscular target, the ocular motoneurons are capable of expressing a greater amount of VEGF and Flk-1 in their neuronal somas as a compensatory mechanism to keep them protected and in an operational state [151,164]. In summary, the improvement in endogenous VEGF levels highlights autocrine functions, and upregulation of the Flk-1 receptor emphasizes the paracrine functions of VEGF in its neuroprotective effect against degeneration [152].

#### 6.3.2. Administration of Trophic Factors

The axotomy of the oculomotor nerves in the adult, and therefore the loss of synaptic connections with the EOMs target, does not trigger the death of ocular motoneurons [185]. However, alterations in their firing patterns and loss of synaptic inputs are observed, which are reversed by the administration of different neurotrophins [140,186]. Thus, although the survival of adult ocular motoneurons does not depend entirely on the neurotrophic supply, this retrograde signaling is required for the maintenance and regulation of their activity and synaptic properties [187,188,189].

It is known that the administration of trophic factors can rescue ocular motoneurons from death after axotomy in early postnatal stages [142], as well as reversing the effects of axotomy in adults by promoting the restoration of synaptic coverage and recovery of tonic-phasic triggering [140,186]. Furthermore, neurotrophins have been shown to promote recovery of the cholinergic phenotype after injury [142,190,191,192]. Therefore, it can be concluded that the ocular motoneurons are characterized by exhibiting a great neurotrophic dependence during the postnatal and adult stages.

In another series of experiments, the recovery of the electrophysiological characteristics of the axotomized motoneurons was observed after the implantation of neural progenitors at the site of the injury [193]. Those neural progenitors expressed NGF, NT-3, and VEGF, revealing for the first time the possible neuroprotective role of VEGF in the oculomotor system. However, it is known that the lack of neurotrophins does not cause motoneuronal degeneration or muscular paralysis, as it does with VEGF deficiency, producing motor alterations similar to that observed during ALS [5,84,194,195]. Therefore, VEGF is probably the most potent of all the neurotrophic factors tested in experimental ALS models [195].

Several experiments have shown that VEGF administration at the injury site can alleviate motoneuronal degeneration in animal models of ALS [36,38,67,82,196]. Recently, the exogenous application of VEGF after axotomy of the abducens nerve prevented the changes observed in axotomized motoneurons, restoring their electrophysiological, morphological properties, and synaptic coverage [173]. These data are correlated with the recovery of ChAT activity observed in axotomized oculomotor neurons due to the administration of VEGF at the site of injury [164].

All these results, together with those that affirm that VEGF administration at the injury supposes a reduction of retrograde axonal degeneration [80,81] and a decrease in injury size and apoptosis levels [87], make this factor an exciting candidate to restore the effects produced by brain damage.

## 7. Conclusions

In summary, all these data suggest that the higher level of VEGF and its receptor Flk-1 observed in extraocular motoneurons may contribute to their higher resistance shown in adverse conditions, such as excitotoxicity, brain damage, or neurodegenerative diseases, such as ALS. The extraocular motor system presents a series of characteristics that favors the correct contribution of VEGF to ocular motoneurons, even during degeneration and denervation processes, compared to what is observed in other most vulnerable motoneurons. The differential presence of VEGF detected in the soma of the oculomotor and non-oculomotor brainstem motoneurons may be the result of a more generous retrograde trophic contribution of VEGF from the EOMs. Furthermore, the fact that after induction of various types of brain damage, there is an increase in the expression of VEGF and Flk-1 in this specific population of motoneurons further highlights the importance of this neurotrophic factor on motoneuronal survival.

## Figures and Tables

**Figure 1 ijms-22-00814-f001:**
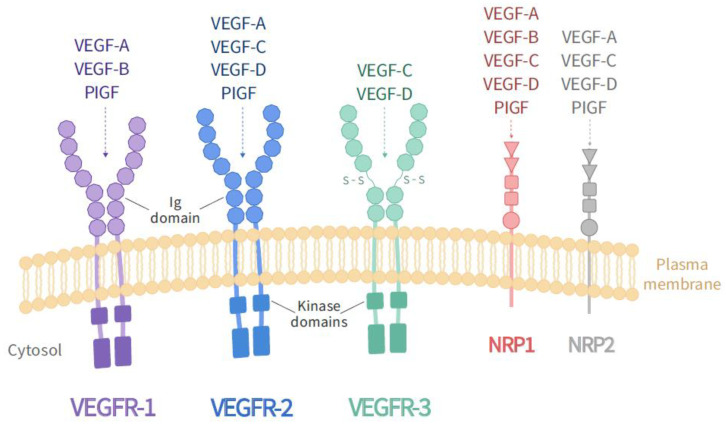
VEGF receptors. The family of VEGF receptors includes three tyrosine kinase receptors (VEGFR-1, VEGFR-2, VEGFR-3) and two non-tyrosine kinase neuropilin receptors (NP-1, NP-2). The different members of the VEGF family bind to the different types of VEGF receptors, as illustrated. The main effect of VEGF-A as a neurotrophic factor is mediated by its binding to VEGFR-2 (Flk-1).

**Figure 2 ijms-22-00814-f002:**
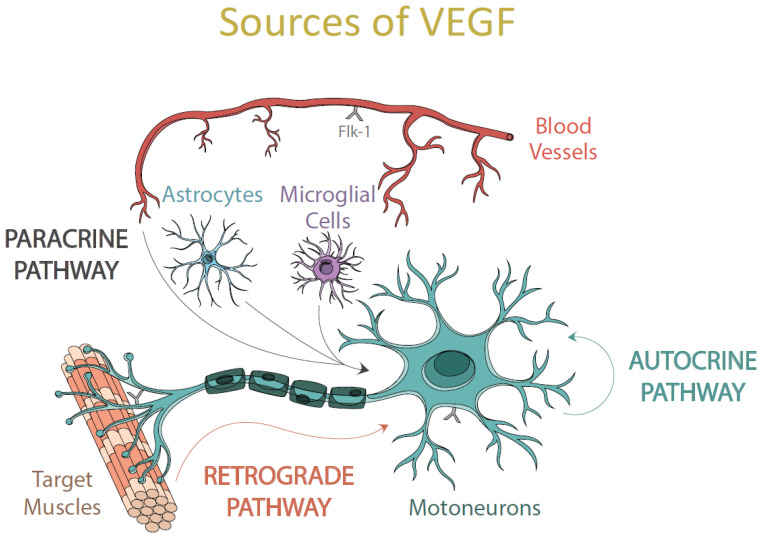
In this scheme, the different pathways of VEGF supply for motoneurons are illustrated. (i) autocrine: self-production of VEGF by the motoneurons themselves; (ii) paracrine: VEGF arriving at the motoneurons from the surrounding cells and blood vessels; and (iii) retrograde: VEGF can also reach the soma of motoneurons from their target muscles.

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
