# Peer review of "Neuroprotective Effect of Vascular Endothelial Growth Factor on Motoneurons of the Oculomotor System"

_ijms, 2021, doi:10.3390/ijms22020814_

Round 1

Reviewer 1 Report

The article by Silva-Hucha et al discussed in addition to angiogenesis, VEGF can protect motoneurons of the oculomotor system. High levels of VEGF and its receptor contribute to the higher resistance of extraocular motoneurons to injury and neurodegenerative diseases. The manuscript is well-written and discusses thoroughly all aspects of VEGF in protecting oculomotoneurons. The manuscript can be accepted for publication in present form.

Author Response

We are very grateful for the reviewer’s comments and for his/her revision of the manuscript.

Reviewer 2 Report

In this review article the authors first give a general introduction to VEGF and its role as a neuroprotective factor in ALS. In the 2nd part of the review article they focus on the role of VEGF as a neurotrophic and neuroprotective factor for ocular motoneurons in ALS and after axotomy. As the group of the authors appears to be the only group worldwide that is working on this subject this 2nd part of the review article is very much focused on the own work of the authors.

In this sense this review article is a mixture of an extensive review of VEGF as a general neuroprotective agent for motoneurons and a more conceptual part advocating the concept that VEGF acting on the Flk1 receptor is a crucial component of the known higher resistance of ocular motoneurons to lesions or neurodegenerative diseases such as ALS.

From my perspective this review article would be more interesting if it was focused more clearly on the conceptual part which is based on the work of the author’s laboratory. The first part with the general review of VEGF as angiogenic and neurotrophic factor could be shortened and the central hypothesis of the authors, i.e. that VEGF activity via the Flk1 receptor is a major factor for neuroprotection and increased resistance of oculomotor neurons should be more in the focus. As this 2nd part is almost entirely based on the authors’ own work the format of a conceptual hypothesis-driven review would be more appropriate than the present format of a more traditional review article on a particular subject given that the vast majority of publications on thesis subject are from the group of the authors.

In this respect, it would be helpful if the authors could discuss in more detail the concepts of the other group working on oculomotor survival (the Hedlund group, e.g. reviewed by Nijssen et al. 2017) and compare them to their own concepts.

Author Response

We really appreciate the suggestions made by the reviewer which have considerably improved the manuscript.

Regarding the introductory part referring to VEGF, the authors consider it essential to introduce the general neuroprotective effect of VEGF on motoneurons to understand more clearly the possible specific protective effect on motoneurons of the oculomotor system, thanks to its more significant presence.

As suggested by the reviewer, a discussion of the work of Dr. Hedlund's research group has been incorporated in section 6 of the review (lines 295-310 and 418). We consider that this discussion has substantially improved the manuscript. Thank you very much for the suggestion made.

Reviewer 3 Report

The authors here have presented a sound and concise review manuscript focussing on the neuroprotective effects of VEGF.

Line 528: The authors contributions need to be amended to take into account that this is a review manuscript and as such no experiments were formed.

Author Response

We are very grateful to the reviewer for his/her review of the manuscript.

We have amended Author Contributions section, eliminating “performance of the different experiments”, as suggested (see lines 540-541).